# Vitamin D May Be Connected with Health-Related Quality of Life in Psoriasis Patients Treated with Biologics

**DOI:** 10.3390/jpm12111857

**Published:** 2022-11-07

**Authors:** Iulia-Alexandra Paliu, Simona-Laura Ianosi, Adina Turcu-Stiolica, Catalina-Gabriela Pisoschi, Luminita-Georgeta Predoi, Andrei-Adrian Tica

**Affiliations:** 1Department of Pharmacology, University of Medicine and Pharmacy of Craiova, 200349 Craiova, Romania; 2Department of Dermatology, University of Medicine and Pharmacy of Craiova, 200349 Craiova, Romania; 3Department of Pharmacoeconomics, University of Medicine and Pharmacy of Craiova, 200349 Craiova, Romania; 4Department of Biochemistry, University of Medicine and Pharmacy of Craiova, 200349 Craiova, Romania; 5Dermatology Ambulatory Office, Clinical Emergency County Hospital of Craiova, 200642 Craiova, Romania

**Keywords:** vitamin D, vitamin D deficiency, biological treatment, psoriasis, health-related quality of life

## Abstract

Suboptimal states of vitamin D may play a role in psoriasis evolution, but the interconnections have been studied over the past years with controversial results. Although a peerless therapy among moderate to severe types of psoriasis, the therapeutic effectiveness of biological therapy may vary unforeseeably between patients and leads to biologics switch. We conducted a pilot study in patients diagnosed with psoriasis and treated with biologics, the purpose of which was to explore the prevalence of suboptimal states of vitamin D, especially in the group of patients characterized by the failure of previous biologics, and to investigate the associations between vitamin D levels and psoriasis, regarding aspects such the severity of the disease and quality of life. Their current result of latent tuberculosis infection (LTBI) was also considered concerning a feasible relationship with vitamin D levels. From July to December 2021, 45 patients corresponding to our inclusion criteria were assessed. Variables such as Psoriasis Area and Severity Index (PASI) score and the Dermatology Life Quality Index (DLQI) score, as well as vitamin D serum concentrations and their LTBI result, were recorded for them. Lower serum concentrations of vitamin D were not more common in patients characterized by failure to previous biologics (*p* = 0.443), but we concluded a weak correlation between the DLQI score and vitamin D (rho = −0.345, *p*-value = 0.020), although a statistically insignificant result was obtained between vitamin D and the PASI score (rho = −0.280, *p*-value = 0.062), and with the LTBI result (rho = −0.053, *p*-value = 0.728). These results establish a connection between higher levels of vitamin D and a better outcome of psoriasis from the perspective of the patient’s quality of life, with no significant association with psoriasis severity and no significant prevalence of suboptimal states among patients that failed previous biologics compared to those with a continuously good response.

## 1. Introduction

Psoriasis is an inflammatory disease of the skin whose prevalence varies between 0.51% to 11.43% in adults and from 0% to 1.37% in children worldwide [1]. In view of pathophysiology, psoriasis involves a cellular immune response mediated by type 1 (T_H_1) and type 17 (T_H_17) T lymphocytes activated by cytokines secreted by antigen-presenting cells in the skin, leading to chronic inflammation and hyperproliferation of keratinocytes [2,3]. Also, a genetic component exists in its etiology, with many genome-wide association studies that indicate more than 80 loci associated with the risk of psoriasis development, the most important being PSORS1 with its susceptible allele HLA-Cw6 [4].

In studies conducted over the past years, there has been an association between low levels of vitamin D and the possibility of psoriasis onset [5]. The involvement of vitamin D in this immune-mediated pathology may be explained by its immunomodulatory properties that prevent autoimmune responses against self-structures, suppressing the expression of histocompatibility major complex class II, lymphocyte T proliferation, pro-inflammatory cytokine secretion, and, alternatively, acting on the production of interleukin (IL)-10, an anti-inflammatory cytokine [5,6,7].

The connection between vitamin D and psoriasis outcome remains, however, controversial and understudied, with variable results recorded on the Psoriasis Area and Severity Index (PASI) score and a dearth of information on the Dermatology Life Quality Index (DLQI) score [8,9].

For moderate to severe types of psoriasis, biological therapy has become a common practice in the last decades. The mechanism of action for biologics used in psoriasis treatment is targeting different cytokines such as tumor necrosis factor (TNF)-α, IL-12/23, IL-17, and IL-23. The studies performed in real-life conditions are mandatory for understanding safety concerns or data that provide efficacy results for psoriasis therapy, especially considering the novelty of some biological agents, significant importance being assigned for the studies including elderly-population or patients with comorbidities including opportunistic infections such as latent tuberculosis infection (LTBI) [10,11,12,13].

Although the discovery of biological drugs like monoclonal antibodies served as a paradigm in the treatment of moderate to severe forms of psoriasis, the therapeutic response may vary between different patients, with a decrease of efficacy in some groups that leads to biological treatment switch [14]. Moreover, the conclusions obtained from real-life studies indicate that once a primary failure to respond to one of the biologics is reported, that may be predictive of the subsequent loss of efficacy [10].

Owing to the suppressing effect of vitamin D on pro-inflammatory cytokines, the assumption of possible connections between vitamin D and biologics became relevant. To date, few studies have looked at the relationship between biological therapy and vitamin D pathways [15,16,17,18], but in a retrospective study of patients with inflammatory bowel disease (IBD) treated with biologics, vitamin D insufficiency was correlated with the earlier cessation of anti-TNF-α drugs due to the failure of the therapy, especially in patients not exposed to pharmaceutical supplementation [19].

On account of the immunosuppressive effects of biologics, at the present time, the current European Guidelines (EuroGuiDerm) recommend screening and rescreening for tuberculosis (TB) according to regional regulations and risk exposure considerations [20]. Taking into consideration the highest prevalence of TB in Romania, the country where our study took place, among European countries, and the prevalence of 6.378 patients with active TB in Dolj county registered in a study performed in 2018 [21], in order to avoid opportunistic infections reactivation or “de novo” infection, the importance of annual TB screening demonstrated by a 9-year retrospective study [22] should not be neglected.

In terms of vitamin D connections, there is evidence that a lower serum level of the metabolite 25-OH vitamin D correlates with a higher chance of LTBI infection, probably due to the functions exercised by vitamin D in the functions of the immune system that may play an important role in the host’s defense against TB [23,24].

Considering all of the above, the purpose of our study is to investigate if lower levels of vitamin D are more common among patients characterized by switching biologics due to ineffectiveness compared to those with a continuously good therapeutic response, and also we aim to analyze the possible associations between vitamin D and psoriasis evolution described by specific scores such as PASI and DLQI. Assessment of the LTBI screening performed under described considerations was registered for statistical analysis as well.

Lifestyle factors consisting of vitamin D- rich foods intake and sun habits, and demographic characteristics were also considered in terms of comparison between different features and daily practices as a possible explanation for a suboptimal state.

## 2. Materials and Methods

### 2.1. Selection and Distribution of Patients into Groups

Our study has received approval from the Ethics Committee of The University of Medicine and Pharmacy of Craiova, Romania (Nr. 54/20.04.2021), and was conducted in accordance with the Declaration of Helsinki, all patients being required to sign an informed consent.

According to our inclusion-exclusion criteria, 45 patients, of both genders, with moderate to severe plaque psoriasis were assessed in the period July–December 2021 in the Dermatology Department of the Clinical Emergency County Hospital of Craiova, Dolj County, Romania.

The inclusion criteria were: age above 18 years, a moderate to severe type of plaque psoriasis, and current treatment with biologics-monoclonal-antibodies (anti-cytokine agents).

The exclusion criteria were: exposure to phototherapy, drugs, and diseases known to affect vitamin D absorption, metabolism, or excretion-concomitant treatment with cytochrome P450 inductors (antiepileptic drugs, rifampicin) or inhibitors (antifungal azoles, antiretroviral), systemic corticosteroid drugs, or biliary cirrhosis, exocrine pancreas insufficiency, gluten enteropathy, short-bowel syndrome, nephrotic syndrome, primary hyperparathyroidism [25]. We also excluded patients exposed to pharmaceutical vitamin D in the previous 6 months or patients with poor adherence to biological therapy. Moreover, since the evidence is lacking about the influence of the available commercial preparations containing a combination of a topical vitamin D derivative, calcipotriol, and a topical glucocorticoid on vitamin D levels, their systemic absorption could not be completely excepted [26], and we also excluded the patients treated with this kind of product.

We divided the patients into 2 categories, as we described in Figure 1, according to the history of biological therapy: Group 1 included the patients who needed to switch the biological drug due to the lack of efficacy, and Group 2 included the patients with a continuously good therapeutically response, without any history of switching biologics. The differences in the exposure times to biologics between the 2 groups are statistically insignificant (*p*-value = 0.297), and therefore, it allows us to compare them.

The lack of efficacy was described as patients with ≤50% reduction in disease severity (calculated by PASI score) and without an appropriate improvement in physical, social, or psychological functioning described with ≤5-point improvement in DLQI score, or patients who lose their initial satisfactory response afterward [27].

### 2.2. Measurement of Vitamin D and Other Variables

For the quantitative determination of vitamin D serum concentrations, the total 25-OH vitamin D was measured in serum using Vitamin D total reagent (from Roche Diagnostics) on the platform Cobas e601 by electrochemiluminescence immunoassay at the Laboratory of the Clinical Emergency Hospital of Craiova. Our assessment consisted of 1 measurement of the serum vitamin D performed at the patients’ routine evaluation in the gap period between the administered doses.

According to the current guidelines [25], the results of the vitamin D serum concentrations obtained were interpreted as normal vitamin D levels > 30 ng/mL, insufficient vitamin D levels between 20–30 ng/mL, and vitamin D hypovitaminosis < 20 ng/mL. Suboptimal states were marked as hypovitaminosis and insufficiency, corresponding with a serum concentration of vitamin D below 30 ng/mL.

Considering the effect of season variation on vitamin D [28], before we performed the statistical analysis, we ensured that the distinctions registered for patients assessed in separate months would not influence our research. So that we did not record a significant difference: medium serum concentration of 27.63 ± 11.11 mg/dl 25-OH vitamin D for patients assessed between July-September and 33.41 ± 13.05 mg/dl for those assessed between October–December.

Patients undergoing biological therapy were screened with the interferon-γ release assay (IGRA)-quantiFERON-TB Gold (QFT) test annually: the test was considered positive in the case of interferon released by T helper lymphocytes CD4+ and T cytotoxic lymphocytes CD8+ at Mycobacterium tuberculosis antigen stimulation > 0.35 UI/mL.

A positive IGRA result was conclusive for LTBI, as none of our patients had active TB during the study period based on their clinical features and findings on chest radiographs. The results included in this study correspond to the values recorded at the time of blood collection for vitamin D assessment.

Usual blood tests were also processed, and we included erythrocyte sedimentation rate (ESR) in our statistics as a usual marker of inflammation.

PASI and DLQI scores were calculated by a dermatologist applying the scoring formula for each of them after evaluating the patient.

PASI score is used to evaluate severity and treatment response. It analyzes the erythema (E), infiltration (I), and desquamation (D) of the disease by giving from 0 (no involvement) to 4 points (severe involvement). The body must be divided into 4 regions: head (h), trunk (t), upper extremities (u), and lower extremities (l), and for each region, a score from 0 (no involvement) to 6 points (>90% involvement) is given for affected surface area (A). The total score ranges from 0 to 72 points and is calculated by this formula:PASI = 0.1 ∗ (E_h_ + I_h_ + D_h_) ∗ A_h_ + 0.2 ∗ (E_t_ + I_t_ + D_t_) ∗ A_t_ + 0.3 ∗ (E_u_ + I_u_ + D_u_) ∗ A_u_ + 0.4 ∗ (E_l_ + I_l_ + D_l_) ∗ A_l_

PASI score for moderate to severe psoriasis sums > 10 points without treatment, but under an effective treatment, a total reduction of at least 75% of PASI must be achieved [29].

A total reduction of PASI with ≥50% but <75% will need a decision based on the DLQI score [27].

DLQI is a scoring questionnaire used for assessing dermatology-specific health-related quality of life (HRQoL). It consists of self-reported perceptions of the patient on 10 questions regarding daily activities, work, school, leisure, personal relationships, treatment, symptoms, and feelings. All questions concern activities from “last week”.

The score ranges from 0 to 30 points, and it is calculated by giving 0 (not at all) to 3 (very much) points to each question. For moderate to severe psoriasis, a score of DLQI > 10 is expected without proper treatment, and a reduction to DLQI ≤ 5 should be achieved for a claim of successful treatment [30,31].

The survey of lifestyle factors consisted of a self-administered questionnaire concerning daily sun habits, food intake, and the use of sun protection factor (SPF) products.

For sun habits, the options for daily sun exposure include “less than 2 h”, “between 2–4 h”, “between 4–6 h”, or “more than 6 h”.

A “yes” or “no” answer was available for the questions concerning the regularity of usage of an SPF of at least 30 with each exposure to the sun. SPF products usually consist of cosmetics such as creams, lotions, sprays, or sticks.

Food intake refers to the frequency (daily, weekly: at least 3 days per week but not a daily practice, or infrequent: less than three 3 per week) of consumption of vitamin D foods, including fish, egg yolk, liver, and dairy [32].

### 2.3. Statistical Analysis

Mann-Whitney U test was used to analyze continuous variables. The χ2 test or Fisher’s exact test were used to compare categorical variables when appropriate. The Spearman correlation coefficient, together with the heatmap matrix, were analyzed for correlations between the outcomes. All statistical analysis was conducted with GraphPad PRISM 9.3.1 software (GraphPad Software, San Diego, CA, USA). The power analysis for our study was performed using G*Power 3.1.9.7 at a 95% confidence level and power factor of 80% for each of the groups. A two-sided *p*-value less than 0.05 was statistically significant. The power test was done, and assuming an alpha level of 0.05, the patients from Group 1 and Group 2 yielded a power between 75% and 88% for the different analyses.

## 3. Results

### 3.1. The Analysis of Demographic Characteristics and Disease-Related Aspects

The study population included 45 adult patients of both genders (60% male) with moderate to severe psoriasis. The division into two groups was made according to biological treatment records as we described in the Section 2.

Table 1 summarizes the characteristics of the two studied patient groups divided according to biological treatment records. Results of vitamin D serum concentration were also interpreted and presented.

According to our analysis, there were no significant differences between the two groups considering demographic characteristics such as age, gender, or residence.

For vitamin D suboptimal serum concentrations, we recorded a percentage of 65% of patients in Group 1 and 48% in Group 2 compared to a percentage of 35% in Group 1 and 52% in Group 2 recorded for normal states.

Although there is a tendency for a higher percentage of patients with lower levels of vitamin D in Group 1, the group of patients characterized by switching of prior biologics due to ineffectiveness, compared to Group 2, the statistical analysis for these results leads to an insignificant *p*-value of 0.443.

Excepting the results of higher vitamin D serum concentrations acquired in men compared with women (*p*-value = 0.036) and in rural citizens compared with urban citizens (*p*-value = 0.008), for vitamin D results, there were no other significant differences according to age characteristics, including elderly distribution, plaque psoriasis type (I or II), gender, residence, BMI (body mass index), and ESR, as all the obtained *p*-values were insignificant. ESR values were higher in female patients (*p*-value = 0.001).

IGRA result was also analyzed, and the results obtained between current positivity and lower vitamin D serum concentrations are statistically insignificant (rho = −0.053, *p* = 0.728).

### 3.2. The Analysis of Biological Treatment Evolution Regarding Vitamin D Aspects

The drug of choice had an influence on patient’s membership to one of the groups (*p*-value = 0.01), as we presented in Table 2, with Etanercept being registered as a treatment in 36% of our Group 2 patients, as well as the drug class (*p*-value < 0.0001), the TNF’s inhibitors corresponding as first-line drugs in 90% of our patients divided in Group 2.

The TNF-α inhibitors are represented by Etanercept, Adalimumab, and Infliximab, while IL-inhibitors used in our patients’ treatment are Ustekinumab (anti-IL-12/23), Secukinumab, Ixekizumab (anti-IL-17), and Risankizumab (anti-IL-23).

However, the variability of the use of different classes of biologics and biological agents does not seem to be associated in any kind with vitamin D results (*p*-value = 0.808).

Our results suggest that the associations between vitamin D levels and psoriasis scores deserve close observation. We concluded that a higher serum level of vitamin D was weakly correlated with a lower DLQI score (rho = −0.345, *p*-value = 0.020), meaning that the higher the vitamin D level is, the lower will be the DLQI score; however, for the PASI score, although there seems to be a tendency towards a negative correlation, we could not prove a significant result (rho = −0.28, *p*-value = 0.062).

According to our linear regression model, starting with vitamin D levels of 32.2 ng/mL could improve patients’ DLQI.

Higher PASI and DLQI scores were recorded in the switched group, Group 1, compared to results registered in Group 2 (*p*-value = 0.018 for PASI, *p*-value = 0.021 for DLQI), and there is also a direct positive correlation between PASI and DLQI score (rho = 0.921, *p*-value < 0.0001).

### 3.3. Factors That Might Influence the Vitamin D Result

In order to identify the differences between daily habits or demographic characteristics that may affect vitamin D concentrations, based on our patients’ statements, we analyzed the connections between lifestyle factors, described as intake of vitamin D-rich foods and sun habits, and vitamin D serum concentrations results, along with the other co-variables presented before.

Spearman’s correlation analysis revealed a significant negative correlation between residence and sun exposure (*p*-value = 0.003), but a direct correlation between vitamin D and sun exposure itself could not be proved (*p*-value = 0.184), although the patients living in the rural zone had a prolonged sun-exposure (*p*-value = 0.003).

The strength of the characteristics between the outcomes of the patients was represented by the color of the square at the intersection of the outcomes, as in Figure 2, Colors range from bright orange (strong positive correlation; ρ = 1.0) to bright green (strong negative correlation; ρ = −1.0). Gender (0 = male, 1 = female) was also found to be negatively correlated with vitamin D (ρ = 0.316, *p*-value = 0.034) and sun exposure (ρ = −0.389, *p*-value = 0.008).

Dairy products were by far the most reported category of consumption from the list of vitamin D foods (approximately 90%). Even so, most of the patients, 45% in Group 1 and 60% in Group 2 reported a frequency of no more than 3 days per week of dairy products consumed, and only 15% and 20%, respectively, claimed a daily intake.

Also, the use of Sun protection factor (SPF) products was not a common practice among our groups of patients involved, although we strongly highlight the importance of daily usage of these products in order to avoid burns or skin cancer. The *p*-values were statistically insignificant for both co-variables dietary intake and SPF use in correlation with vitamin D (*p* = 0.570, *p* = 0.166).

## 4. Discussion

Despite the evidence obtained from studies performed in patients with IBD undergoing biological therapy, the connections between biologics and vitamin D are lacking for psoriasis patients. Considering the research background, patients with IBD with vitamin D insufficiency had shorter durability for the biological treatment with TNF-α inhibitors. [19,33].

In our study of psoriasis patients, the prevalence of suboptimal states, marked as hypovitaminosis and insufficiency, in the two groups was a little higher in Group 1 (65%) compared to Group 2 (48%), but the situation was reversed when we analyzed the situation of normal vitamin D serum concentration (52% of patients in Group 2 compared to 35% in Group 1). But although the tendency exists based on the percentage distribution, the *p*-value was statistically insignificant in terms of comparison between groups’ concentrations of vitamin D.

We must note that there may be a possible reason for dissimilarities between IBD studies and studies performed on psoriasis as a more severe form of IBD is observed in patients with lower levels of vitamin D, and therefore greater demands for the therapy are needed [34], but in the case of psoriasis, there is no current evidence to support the same observation.

Evidence provided by a study performed in psoriasis patients suggests that the use of TNF-α inhibitors leads to lower concentrations of the vitamin D [18], but another study focused only on Etanercept administration did not prove any link between lower levels of vitamin D and the use of the biological drug [15].

In our research, current treatment with Etanercept was the preferred agent for the first line of therapy, with 36% of patients in Group 2, the non-switched group, with this biological agent as their current treatment.

Considering all of the above, in our study, Etanercept, a TNF-α blocking agent, does not seem to be more associated with the suboptimal states of vitamin D, the comparison with the normal state being insignificant (20% vs. 16%) for Group 2. The situation is also similar in Group 1, with a tie between deficient states (5%) and normal states (5%).

Moreover, none of the biologics, neither a TNF-α blocking drug nor an IL inhibitor, could be assigned to a significant association with vitamin D levels (*p*-value = 0.808). However, we want to highlight that our expertise in the aspect of causality between exposure to biologics and vitamin D deficiency is limited by the cross-sectional design of the study, our study being only able to provide association features based on present records. Also, our study consisted of only one measurement of vitamin D level performed during the gap period between administered doses.

During the treatment with biological drugs, concomitant LTBI is a well-known contraindication for TNF-α treatment initiation; the current guidelines indicate chemoprophylaxis for at least 1 month in cases of LTBI positivity [20]. Currently, anti-IL-17 and anti-IL-23 biologics did not raise concerns in terms of LTBI reactivation [20,35]. However, the risk for “de novo” active tuberculosis still exists [36], and in view of the high burden of TB in our country, as a strategy to help prevent the spread of the disease, annual screening is still recommended for all categories of patients undergoing biologics.

The significance of these records for our study is based on the evidence that lower levels of circulating vitamin D were correlated with a higher chance of LTBI [23], but despite the evidence, in our research, we could not find any link between vitamin D results and current IGRA positivity.

As for connections between psoriasis and vitamin D, our study wants to highlight the following aspects:

First of all, there is a remarkable difference between studies that assess the connections between psoriasis and circulating levels of vitamin D at its basal levels and those that investigate the role of pharmaceutical supplementation in view of a better outcome of the disease.

A meta-analysis from 12 different studies evaluating circulating levels of vitamin D in psoriasis patients claimed a small but significant inverse correlation between low levels of vitamin D and psoriasis severity described by PASI score [8], but the reports on DLQI score are lacking.

On the other hand, pharmaceutical supplementation of vitamin D offers contradictory results for the improvement of the disease. One of the reasons for conflicting results achieved could be allocated to different doses regimen. The doses used for supplementation between 1986 and 2013 were comprised of between 0.25–2.0 µg/day, doses up to 1500 µg (60000 IU) given every 2 weeks for 6 months, being mentioned in clinical trials only after 2014 [37]. Second, even if there is evidence of some studies claiming benefits in oral administration of vitamin D, the improvement of PASI score could not be proved by all clinical trials performed even with high doses of 100 µg/day (4000 IU) [37,38,39,40]. To our knowledge, only Jarrett et al. included the DLQI score in a clinical trial with pharmaceutical supplementation of vitamin D as a tool of evaluation, but without achieving a significant result [40].

In our study, measurement of vitamin D level was made without any pharmaceutical supplementation, but a statistically significant correlation with PASI score could not be proved, although there is a tendency for the result to be notable (*p*-value = 0.062).

Instead, the relationship between vitamin D and DLQI score has a weak negative correlation (rho = −0.345, *p*-value = 0.020), establishing a connection between suboptimal states of vitamin D and impaired quality of life. We also concluded that according to our analysis, the necessary values for the improvement of the DLQI score starts with the levels of 32.2 ng/mL, this result indicating a value of vitamin D above the suboptimal range.

The importance of this result could be a major consideration for future studies targeting psoriasis patients’ well-being as the disease impact on Health-Related Quality of Life (HRQoL) scales could be commensurate to diseases like cancer, arthritis, or depression. DLQI was the first skin-related score that assessed HRQoL, and considering that a reduction in PASI score, without a reduction in DLQI score, does not meet the therapeutic goals, its importance should not be neglected [27,30,41]

A possible explanation for our result, associating vitamin D with DLQI score, could be assigned to the interrelation between low vitamin D concentration and depressive features [42,43], knowing that depressive disorder has already been linked to a negative impact on HRQoL in psoriasis patients according to their DLQI score and the brain-skin axis influence the clinical course of the disease by dysregulation of different components of the nervous or immune system through the secretion and action of several proinflammatory mediators [44,45].

These dissimilarities between studies that research the connection between psoriasis and vitamin D at its basal circulating level and those that investigate the outcome of pharmaceutical supplementation may be explained by the natural variability in the key metabolic enzymes of vitamin D: 25-hydroxylase (CYP2R1) and 1-α-hydroxylase (CYP27B1) and vitamin D receptor (VDR) that can affect the efficacy of supplements [46], but we strongly believe that further research would be needed for other possible unknown causes.

For factors that could influence vitamin D levels in terms of different daily habits or different demographic characteristics, we obtained the following results:

According to gender differences, our research found higher vitamin D serum levels in men patients compared to women (*p*-value = 0.036), as other studies demonstrated [47,48]. Also, the male gender seems to have different sun habits, with longer reported exposure to the sun (*p*-value = 0.034).

For recording a normal vitamin D serum concentration, lifestyle factors may play an important role. UV-B radiation from sun exposure is the main natural source of vitamin D [49]. Studies performed in tropical and South-East Asian countries reported a higher prevalence of vitamin D hypovitaminosis in urban citizens [50,51,52,53]. However, despite all the evidence, the connection between these pieces of information is not completely understood, as data recorded on rural agricultural workers who spend up to eight hours of sun exposure per day claimed a prevalence of vitamin D hypovitaminosis of 44% in men and 70% in women [48] Considering this, we also registered a positive correlation between residence and vitamin D, with higher serum concentrations in the rural region, but we could not find a direct correlation between vitamin D and sun exposure itself (*p*-value = 0.184), although the patients living in the rural zone had a prolonged sun-exposure (*p*-value = 0.008). Furthermore, the correlations between higher vitamin D serum concentrations in men and also, prolonged reported exposure of them to the sun would theoretically suggest a connection, but the insignificant *p*-value obtained for the analysis of the interrelationship between vitamin D and sun exposure, makes us believe that a more complex pathway is involved.

In Romania, the dietary intake of vitamin D sources is inadequate, and the fortification of food with vitamin D is not a current practice [54]. In dairy products, the natural vitamin D amount is 0.1 µg/100 g in whole milk and yogurt and 0.3–0.6 µg/100 g in cheese [55]. As the values for vitamin D intake in order to avoid deficiency or insufficiency reach 10 µg/day, respectively 26 µg/day [56], food intake was not a reason for suboptimal states in our patients, considering the greater preference recorded by our survey for the dairy category as the main source of dietary vitamin D, we emphasize the need for awareness of the nutritional deficit in order to prevent suboptimal states of vitamin D in the future.

## 5. Conclusions

In our research, suboptimal states of vitamin D do not characterize in a significant manner the group of patients with failure to previous biologics in comparison with those with a continuously good therapeutic response, and also, lower concentrations of vitamin D could not be assigned to current LTBI positivity. Instead, in our groups of patients treated with biologics, a higher serum concentration of vitamin D seems to be associated with a better outcome of psoriasis in terms of health-related quality of life leading to a whole new perception of its involvement in patients’ well-being. However, the result is insignificant for the relationship between vitamin D and psoriasis severity, whereas our sample considered patients non-exposed to pharmaceutical supplements of vitamin D.

## Figures and Tables

**Figure 1 jpm-12-01857-f001:**
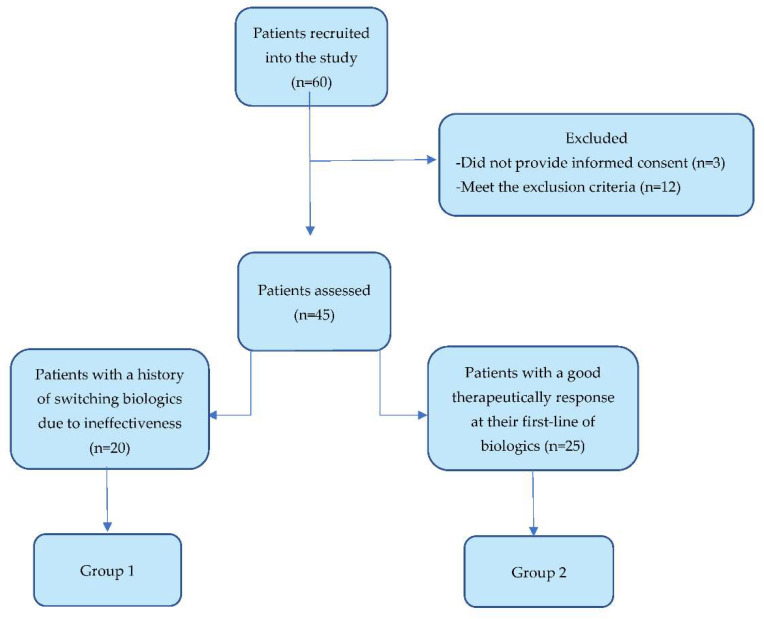
Flowchart of patients included in the study and distribution criteria for the two groups.

**Figure 2 jpm-12-01857-f002:**
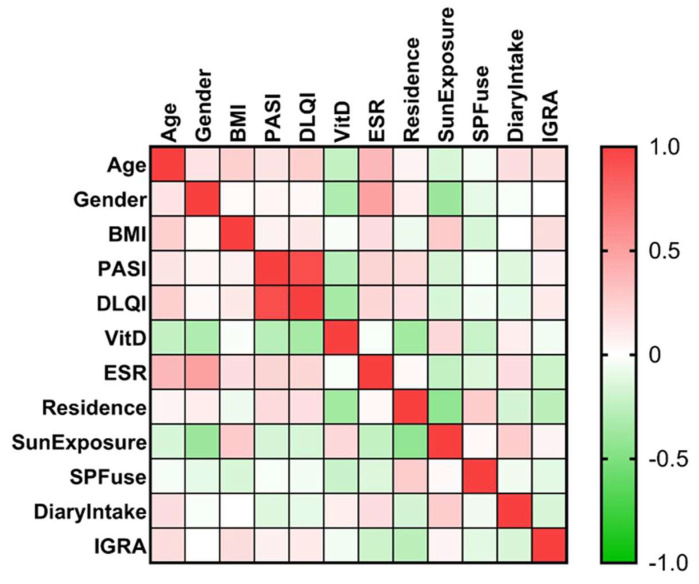
Heatmap of the correlation matrix.

**Table 1 jpm-12-01857-t001:** General characteristics of patients included in the two psoriasis groups.

Mean ± SD Median (Interquartile Range) Range	Group 1	Group 2	*p*-Value
Vitamin D serum concentration			0.443
Hypovitaminosis	3 (15%)	4 (16%)	
Insufficiency	10 (50%)	8 (32%)	
Normal	7 (35%)	13 (52%)	
Age category	58.90 ± 11.0	54.88 ± 10.59	
	64.50 (48.75–68.50)	52 (49–65)	0.262
	39–75	29–71	
<60 years	9 (45%)	16 (55%)	0.821
Hypovitaminosis	1 (5%)	1 (4%)	
Insufficiency	4 (20%)	6 (24%)	
Normal	4 (20%)	9 (36%)	
>60 years	11 (51%)	9 (36%)	0.603
Hypovitaminosis	2 (10%)	3 (12%)	
Insufficiency	6 (24%)	3 (12%)	
Normal	3 (15%)	3 (12%)	
Psoriasis onset < 40 years (type I)	10 (50%)	12 (48%)	0.438
Hypovitaminosis	1 (5%)	0 (0%)	
Insufficiency	5 (25%)	5 (20%)	
Normal	4 (20%)	7 (28%)	
Psoriasis onset > 40 years (type II)	10 (50%)	13 (42%)	0.637
Hypovitaminosis	2 (10%)	4 (16%)	
Insufficiency	5 (25%)	4 (16%)	
Normal	3 (15%)	5 (20%)	
Gender			
Female	7 (35%)	11 (44%)	0.557
Hypovitaminosis	1 (5%)	3 (12%)	
Insufficiency	5 (25%)	5 (20%)	
Normal	1 (5%)	3 (12%)	
Male	13 (65%)	14 (56%)	0.603
Hypovitaminosis	2 (10%)	1 (4%)	
Insufficiency	5 (25%)	4 (14%)	
Normal	6 (24%)	9 (36%)	
Residence			
Urban	13 (65%)	12 (48%)	0.368
Hypovitaminosis	2 (10%)	3 (12%)	
Insufficiency	8 (40%)	4 (16%)	
Normal	3 (15%)	5 (20%)	
Rural	7 (35%)	13 (52%)	0.846
Hypovitaminosis	1 (5%)	1 (4%)	
Insufficiency	2 (10%)	5 (20%)	
Normal	4 (20%)	7 (28%)	
BMI (kg/m^2^)	31.05 ± 6.77	30.48 ± 5.2	
	32.7 (25.25–35.25)	30.86 (26.62–34.69)	0.775
	19.05–43.25	19.84–38.48	
>30 kg/m^2^	12	16	0.343
Hypovitaminosis	1 (5%)	3 (12%)	
Insufficiency	7 (35%)	5 (20%)	
Normal	4 (20%)	8 (32%)	
<30 kg/m^2^	8	9	0.755
Hypovitaminosis	2 (10%)	1 (4%)	
Insufficiency	3 (15%)	4 (16%)	
Normal	3 (15%)	4 (16%)	
ESR			0.472
Hypovitaminosis	3 (15%)	5 (20%)	
Insufficiency	10 (50%)	8 (32%)	
Normal	7 (35%)	12 (48%)	
IGRA result			
Positive	9 (45%)	16 (64%)	0.788
Hypovitaminosis	2 (10%)	3 (12%)	
Insufficiency	4 (20%)	6 (24%)	
Normal	3 (15%)	7 (28%)	
Negative	11 (55%)	9 (36%)	0.353
Hypovitaminosis	4 (20%)	5 (20%)	
Insufficiency	6 (30%)	2 (8%)	
Normal	4 (20%)	5 (20%)	

BMI = body mass index, IGRA = interferon-γ release assay, ESR = erythrocyte sedimentation rate.

**Table 2 jpm-12-01857-t002:** Evolution of treatment in view of membership to one of the two groups.

Mean ± SD Median (Interquartile Range) Range	Group 1	Group 2	*p*-Value
Anti-TNFα agent	2 (10%)	16 (64%)	<0.0001 ****
Anti-IL agent	18 (90%)	9 (36%)	
Etanercept	2 (10%)	9 (36%)	0.01 *
Adalimumab	0	6 (24%)	
Infliximab	0	1 (4%)	
Secukinumab	7 (35%)	2 (8%)	
Risankizumab	5 (25%)	1 (4%)	
Ustekinumab	2 (10%)	3 (12%)	
Ixekizumab	4 (20%)	3 (12%)	
Number of patients with hypovitaminosis	3 (15%)	4 (16%)	
Treated with TNF-α inhibitors	1 (5%)	2 (8%)	
Treated with IL-inhibitors	2 (10%)	2 (8%)	
Vitamin D (ng/mL)	18.09 ± 0.75 (17.44–19.15)	15.22 ± 2.67 (11.68–19.92)	
PASI score	5.83 ± 4.69 0–11.50	2.62 ± 2.82 1–11	
DLQI score	6.33 ± 4.10 0.80–8.00	2.20 ± 1.60 1–4	
Number of patients with insufficiency	10 (50%)	9 (36%)	
Treated with TNF-α inhibitors	0%	6 (24%)	
Treated with IL-inhibitors	10 (50%)	3 (12%)	
Vitamin D (ng/mL)	24.29 ± 3.12 (20.04–29.58)	24.21 ± 3.10 (20.58–28.61)	
PASI score	3.51 ± 3.34 0–11.50	3.15 ± 3.89 0–10.20	
DLQI score	3.7 ± 4.05 0–12	2.62 ± 3.56 0–10	
Number of patients with a normal concentration of vitamin D	7 (35%)	12 (48%)	
Treated with TNF-α inhibitors	1 (5%)	8 (32%)	
Treated with IL-inhibitors	6 (30%)	4 (16%)	
Vitamin D (ng/mL)	43.82 ± 6.63 (33.96–50.84)	42.38 ± 10.30 (31.57–63.62)	
PASI score	3.84 ± 4.62 0.50–14.40	0.68 ±1.26 0–4.20	
DLQI score	3 ± 3.85 0–12	0.75 ± 1.36 0–4	

* *p*-value < 0.05; ****, *p*-value < 0.0001.

## Data Availability

The data presented in this study are available on request from the corresponding author. The data are not publicly available due to patients’ privacy rights.

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
