# Peer review of "Vitamin D May Be Connected with Health-Related Quality of Life in Psoriasis Patients Treated with Biologics"

_jpm, 2022, doi:10.3390/jpm12111857_

Round 1

Reviewer 1 Report

I read with great interest this article titled "Vitamin D May Be Connected With Health-Related Quality of Life in Psoriasis Patients Treated with Biologics" by Iulia-Alexandra Paliu  et al.

I have only some minor revisions:

Did you analyze whether patients used topical preparations containing vitamin D derivatives on skin lesions, if so, for how long and whether it could have an impact on the concentration of vitamin D in the serum?

Have you made calculations at what vitamin D levels improve patients' DLQI?

Author Response

We are very grateful for the constructive comments from the reviewers. 
We have carefully addressed point-by-point all the comments and made corrections in our manuscript using tracked changes.

  • For the first comment, concerning the impact of vitamin D topical derivatives on circulating vitamin D levels, we claim that this category of patients, using these topical products meet our exclusion criteria. Our decision was made because in our country the only available formulation containing topical vitamin D derivatives also includes a topical glucocorticoid in it and considering the lacking evidence of their administration impact on vitamin D levels, and our criteria for including only patients non-exposed to pharmaceutical vitamin D or drugs that could affect its natural metabolism, we decided to exclude them. We specified these aspects at the beginning of the Material and Methods section, and thanks to the reviewer’s suggestion, we modified it more conclusively:

The exclusion criteria were exposure to phototherapy, drugs, and diseases known to affect vitamin D absorption, metabolism, or excretion- concomitant treat-ment with cytochrome P450 inductors (antiepileptic drugs, rifampicin) or inhibitors (antifungal azoles, antiretroviral), systemic corticosteroid drugs, or topical use of calcipotriol, biliary cirrhosis, exocrine pancreas insufficiency, gluten enteropathy, short-bowel syndrome, nephrotic syndrome, primary hyperparathyroidism [25]. We also excluded patients exposed to pharmaceutical vitamin D in the past six months or patients with poor adherence to biological therapy. Moreover, since evidence is lacking about the influence of the available commercial preparations containing a combination between a topical vitamin D derivative, calcipotriol, and a topical glucocorticoid on vitamin D levels, and their systemic absorption could not be completely excepted [26], we also excluded the patients treated with this kind of products.

  1. https://www.ema.europa.eu/en/medicines/human/referrals/daivobet. Annex I list of the names, pharmaceutical form(s), strength(s) of the medicinal product(s), route(s) of administration, applicant(s) marketing authorisation holder(s) in the member states. (accessed on 27 October 2022).

  • For the second comment, we performed a linear regression model analysis, and we concluded that the value of vitamin D corresponding to the improvement in the patients’ DLQI start with the value of 32.2 ng/ml. We specified this in the Results section of the manuscript:

According to our linear regression model, starting with vitamin D levels of 32.2 ng/ml could improve patients’ DLQI.

And in the Discussion section:

We also concluded that according to our analysis, the necessary values for the improvement of the DLQI score starts with the levels of 32.2 ng/ml, this result strongly indicating a value of vitamin D above the suboptimal range. 

Reviewer 2 Report

The manuscript provides an interesting information of relationship among vitamin D and psoriasis, the effects of biologics, and the patient's lifestyle. It is generally well written and reviewer will accept the manuscript for publication.

Pleases modified a few minor concerns.

It is not specified when vitamin D was measured during the of the patient's clinical course. For example, before or after administration of a biologic?  

In addition, is vitamin D only measured once?  

Please specify the timing and number of measurements in the manuscript. If there are limitations to this research, please add them to 4. Discussion.

Author Response

We are very grateful for the constructive comments from the reviewers. 
We have carefully addressed point-by-point all the comments and made corrections in our manuscript using tracked changes.

In order to improve the clarity of the paper, some changes were made to the content of the paper, as follows:

  • Considering the two recommendations, we included in our manuscript, in the Discussion section, the following aspects concerning a potential limitation:

Our assessment consisted of one measurement of the serum vitamin D performed at the patients’ routine evaluation, in the gap period between administered doses.

Also, we specified these in the Material and Method section.